# The Effects of Spirulina Supplementation on Redox Status and Performance Following a Muscle Damaging Protocol

**DOI:** 10.3390/ijms22073559

**Published:** 2021-03-30

**Authors:** Aggelos Pappas, Athanasios Tsiokanos, Ioannis G. Fatouros, Athanasios Poulios, Dimitris Kouretas, Nikos Goutzourelas, Giannis Giakas, Athanasios Z. Jamurtas

**Affiliations:** 1Department of Physical Education and Sport Science, University of Thessaly, 42100 Trikala, Greece; apappa66@yahoo.com (A.P.); atsiokan@pe.uth.gr (A.T.); ifatouros@uth.gr (I.G.F.); athanpoul@gmail.com (A.P.); ggiakas@uth.gr (G.G.); 2Department of Biochemistry and Biotechnology, University of Thessaly, 41500 Larissa, Greece; dkouret@uth.gr (D.K.); nikgkoutz@gmail.com (N.G.)

**Keywords:** exercise, free radicals, nutrition, eccentric, inflammation

## Abstract

Spirulina plantensis is a popular supplement which has been shown to have antioxidant and performance enhancing properties. The purpose of this study was to evaluate the effects of spirulina supplementation on (a) redox status (b) muscle performance and (c) muscle damage following an eccentric bout of exercise that would induce muscle damage. Twenty-four healthy, recreationally trained males participated in the study and were randomly separated into two groups: a spirulina supplementation (6 g per day) and a placebo group. Both groups performed an eccentric bout of exercise consisting of 5 sets and 15 maximum reps per set. Blood was collected at 24, 48, 72 and 96 h after the bout and total antioxidant capacity (TAC) and protein carbonyls (PC) were assessed in plasma. Delayed onset muscle soreness (DOMS) was also assessed at the same aforementioned time points. Eccentric peak torque (EPT) was evaluated immediately after exercise, as well as at 24, 48, 72 and 96 h post exercise. Redox status indices (TAC and PC) did not change significantly at any time point post exercise. DOMS increased significantly 24 h post exercise and remained elevated until 72 h and 96 h post exercise for the placebo and spirulina group, respectively. EPT decreased significantly and immediately post exercise and remained significantly lower compared to baseline until 72 h post exercise. No significant differences between groups were found for DOMS and EPT. These results indicate that spirulina supplementation following a muscle damaging protocol does not confer beneficial effects on redox status, muscle performance or damage.

## 1. Introduction

Spirulina platensis is a cyanobacterium that is rich in protein, essential fatty acids, minerals and vitamins like vitamin C, vitamin E, B12 [1,2,3]. Spirulina has been shown to boost the immune system by increasing the activity of NK cells, interferon production and to protect from viral infections [4]. Furthermore, the anti-inflammatory properties of spirulina have been related to decrements in the release of histamine [4] and daily supplementation with 1 mg of spirulina has been shown to increase the activity of macrophages by 28–39% compared to 24–25% with placebo [5].

Spirulina also possesses antioxidant potential [6] since it contains many compounds like tocopherols, beta-carotenes, phycocyanins [7] that have antioxidant activity which reduce the action of cyclo-oxygenase-2 [8] and it has been reported that supplementation with spirulina and ginseng can increase the reduced levels of reduced glutathione (GSH), a very potent antioxidant compound, the enzymes glutathione peroxidase and superoxide dismutase and decrease lipid peroxidation [9]. Furthermore, supplementation with spirulina in elderly male and female subjects improved their lipid profile, immune variables, and antioxidant capacity [10].

The chronic exercise-induced health benefits are well-known. It is also known that aerobic and anaerobic exercise training are beneficial to improve redox balance in humans, and any type of exercise training will be beneficial against potential risk factors of excessive reactive oxygen species (ROS)-mediated diseases [11,12]. However, acute physical exercise with a certain high intensity and duration may induce an increase in the production of ROS [13,14]. Previous work on spirulina supplementation and exercise training has revealed positive results [15,16,17] and work from our laboratory has shown that acute supplementation of spirulina increased aerobic exercise performance, as was evident by increased time to exhaustion to a determined load, decreased carbohydrate and increased fat oxidation and lower lipid peroxidation [18].

Unaccustomed eccentric exercise has been shown to result in ultrastructural muscular disruption, delayed onset muscle soreness (DOMS), enhancement of oxidative stress and reductions in performance related variables, i.e., impaired muscle force production [19,20]. Furthermore, reports indicate that an alteration of the immune system is evident with neutrophils infiltration at the damaged tissue site taking place in order to clean the necrotic myofibers and clean the cellular debris, a process that oxidative stress is involved in [21]. Increased muscle soreness and losses in muscle function are important for an athlete since in can result in attenuation in exercise performance. Several nutritional interventions have been proposed in the literature to ameliorate DOMS and speed up the recovery of muscle function [22,23]. This is extremely important for athletes participating in a tournament scenario or who participate in a congested game schedule. Therefore, assessing the impact of nutritional supplements like spirulina on muscle soreness and muscle function is of great importance to the athletic population [24]. The majority of research examined the spirulina supplementation effects on aerobic exercise performance and there is no literature assessing the effects of supplementation on anaerobic exercise and more specifically on muscle damage following intense exercise.

Therefore, the purpose of this study was to assess the effects of spirulina supplementation on exercise performance, muscle damage and redox status following an acute eccentric exercise bout.

## 2. Materials and Methods

### 2.1. Participants

A power analysis was conducted by using the G*Power version 3.1.9.2 program. It was estimated that a total sample of 24 participants was necessary to identify statistically meaningful trial effects (with a probability error of 0.05, a statistical power of 80% and an effect size of 0.25). Twenty-four healthy, recreationally trained male participants took part in the study and were randomly separated into two groups: a spirulina supplementation (6 g per day, *n* = 12) and a placebo group (*n* = 12). A dietitian was responsible for constructing the capsules. The daily dosage of spirulina (Algae AC, Serres, Greece) was 6 g per day and the capsules (1 g each) were taken three times per day before meals. Spirulina was given in a capsule form. Placebo was also given in a capsule that contained wheat flour and was of the same color and size as the spirulina capsule to avoid identification of the supplement. Subjects were asked whether they could distinguish between supplements and their response was that could not identify the spirulina or placebo supplement. Furthermore, subjects were also asked to return at the end of trial the capsules that were not consumed; there was no subject that returned capsules at the end of the trial. The dosage of 6 g per day was chosen because it is the same as the one used by our group in a previous study [18] and similar to the ones used by relevant human studies (7.5 g per day) [17]. Inclusion criteria for participation in the study were: (1) normal BMI, (2) absence of musculoskeletal injuries for ≥12 months prior to the study, (3) no consumption of drugs or supplements, (4) avoidance of scheduled eccentric exercise training for ≥6 months. Participants completed a medical and supplementation history questionnaire and they were non-smokers and did not take any supplements for at least 6 months before the commencement of the study. Subjects were informed about the risks and benefits of the study and signed a written informed consent. Procedures were in accordance with the 1975 Declaration of Helsinki, as revised in 2000, and approval was granted by Institutional ethics committee (Ref#: 799/2013). No funding was received for the completion of the present study.

### 2.2. Experimental Design

One to two weeks before the commencement of the study, subjects reported to the lab and baseline measurements were performed. Body weight was measured to the nearest 0.5 kg while subjects were barefoot and lightly dressed (Beam Balance 710; Seca, Birmingham, UK) and standing height to the nearest 0.5 cm (Stadiometer 208, Seca, Birmingham, UK). Eccentric peak torque (EPT), for each leg separately, at an angular velocity of 60° s^−1^ of the knee extensor muscles, was measured on an isokinetic dynamometer (Cybex, Ronkonkoma, NY). The best of 5 maximal eccentric voluntary contractions were recorded for the estimation of EPT. To ensure that the subjects provided their maximal effort, the measurements were repeated if the difference between the lower and the higher torque values exceeded 10%. A two min rest interval was incorporated between the efforts. Delayed onset of muscle soreness (DOMS) was assessed during squat movement (90° knee flexion), and perceived soreness was rated on a scale ranging from 1 (normal) to 10 (very sore). Instructions were given to participants to avoid strenuous physical activity for 3 days prior to experimental protocol and for the whole period of data collection.

### 2.3. Eccentric Exercise Protocol

Prior to eccentric exercise protocol, subjects performed a 10 min warm-up of cycling (70–80 rpm and 50 W) on a Monark cycle ergometer (Monark, Vansbro, Sweden), followed by 5 min of stretching exercises of the major muscle groups of the lower limbs. Eccentric exercise session was performed on an isokinetic dynamometer (Cybex Norm, Ronkonkoma, NY) and exercise protocols were undertaken from the seated position (120° hip angle) with the lateral femoral condyle aligned with the axis of rotation of the dynamometer. Participants were coupled to the dynamometer by an ankle cuff, attached proximal to the lateral malleolus and finally stabilized according to the manufacturer’s instructions. Participants completed 5 sets of 15 eccentric maximal voluntary contractions (knee range, 0° full extension to 90° flexion) at an angular velocity of 60°/s. A 2 min rest interval was used between sets and the total workout time was 15 min. After the end of the eccentric exercise protocol, subjects were provided with the spirulina or placebo capsules and they were obliged to take 6 capsules a day at the previously mentioned time points until the final day of data collection.

### 2.4. Blood Collection

Blood samples (8 mL) were drawn from a forearm vein in ethylenediaminetetraacetic acid (EDTA) tubes at five different time points, namely before exercise, 24, 48, 72 and 96 h post exercise. Samples were centrifuged (1370× *g*, 10 min, 4 °C) and the supernatant (i.e., plasma) was collected and stored in multiple aliquots. Samples were thawed only once before the analysis.

### 2.5. Biochemical Analyses

Protein carbonyls were determined in plasma as previously described [25]. Briefly, 50 μL of 20% trichloroacetic acid (TCA) was added to 50 μL of plasma and this mixture was incubated in an ice bath for 15 min and centrifuged (15,000× *g*, 5 min, 4 °C). The supernatant was discarded and 500 μL of 10 mM 2,4-dinitrophenylhydrazine (in 2.5 N HCl) for the sample, or 500 μL of 2.5 N HCl for the blank, was added in the pellet. The samples were incubated in the dark at room temperature (RT) for 1 h with intermittent vortexing every 15 min and centrifuged (15,000× *g*, 5 min, 4 °C). The supernatant was discarded and 1 mL of 10% TCA was added; the samples were vortexed and centrifuged (15,000× *g*, 5 min, 4 °C). The supernatant was discarded and 1 mL of ethanol–ethyl acetate mixture (1:1 *v*/*v*) was added; the samples were vortexed and centrifuged (15,000× *g*, 5 min, 4 °C). This step was repeated twice. The supernatant was discarded and 1 mL of 5 M urea (pH 2.3) was added; the samples were vortexed and incubated at 37 °C for 15 min. The samples were then centrifuged (15,000× *g*, 3 min, 4 °C) and the absorbance was monitored at 375 nm. Total plasma protein was determined using Bradford method via a standard curve of solutions with known bovine serum albumin concentrations. The intra-assay coefficient of variation for protein carbonyls was 4.6%.

Total antioxidant capacity (TAC) was determined by adding 20 μL of plasma to 480 μL of 10 mM sodium potassium phosphate (pH 7.4) and 500 μL of 0.1 mM 2,2-diphenyl-1-picrylhydrazyl (DPPH) free radical, and the samples were incubated in the dark for 30 min at room temperature. The samples were centrifuged for 3 min at 20,000× *g*, and the absorbance was read at 520 nm. TAC is presented as millimole of 1,1-diphenyl-2-picrylhydrazyl radical (DPPH) reduced to 1,1-diphenyl-2-picrylhydrazine (DPPH:H) by the antioxidants of plasma. The intra-assay coefficient of variation for TAC was 2.6%.

### 2.6. Statistical Analysis

The normality of all dependent variables was assessed by the Shapiro–Wilk test and was found not to differ significantly from normality. A 2-way (group × time) repeated measure analysis of variance (ANOVA) with planned contrasts on different time points was used for data analysis. A Bonferroni test was used for post hoc analysis when a significant effect was detected. The level of significance was set at *p* < 0.05. Data are presented as means ± SD. For effect size determination, partial eta square values (η^2^) were used for repeated measures and were calculated according to a previous study [25]. SPSS version 18.0 was used for all analyses (SPSS Inc., Chicago, IL, USA).

## 3. Results

Personal characteristics of the participants appear in Table 1. Baseline values did not differ between the two groups (*p* > 0.05).

Data analysis did not reveal significant differences (*p* = 0.794; η^2^= 0.012) between the two groups for EPT in both legs (EPT_B_). However, significant time-dependent differences (*p* < 0.05; η^2^ = 0.513) were observed for both groups (Figure 1). A significant decrease was observed for spirulina group immediately after exercise by 29%, at 24 h by 28.7%, at 48 h by 28% and 72 h post exercise by 26.9%; for the placebo group, a decrease was also observed immediately after by 33%, at 24 h by 31.5%, at 48h by 28.8% and 72 h post exercise by 24.9%.

Furthermore, there were no significant differences (*p* = 0.515; η^2^= 0.032) between the two groups for EPT in right leg (EPT_R_). However, significant time-dependent differences (*p* < 0.05; η^2^= 0.326) were observed for both groups (Figure 2). EPT_R_ appeared significantly lower compared to baseline for spirulina group immediately post exercise by 27.8%, at 24 h by 29.1% and at 48 h by 27.5%; for the placebo group, the decrease immediately post exercise was 27.7%. In addition, there were no significant differences (*p* = 0.333; η^2^= 0.05) between the two groups for EPT in left leg (EPT_L_). However, significant time-dependent differences (*p* < 0.05; η^2^ = 0.509) were observed for both groups (Figure 3). EPT_L_ appeared significantly lower compared to baseline immediately post exercise by 30.4%, at 24 h by 28.2%, at 48 h by 28.5% and at 72 h post exercise by 28.8% for spirulina group; for the placebo group, EPT_L_ decreased immediately after exercise by 37.5%, at 24 h by 41.2%, at 48 h by 36% and at 72 h post exercise by 30.3%.

DOMS was significantly (*p* < 0.05; η^2^ = 0.772) increased at 24 h by 220.8%, at 48 h by 298.8%, and at 72 h by 212.5% following the exercise protocol in spirulina group and also at 24 h by 237.7%, 48 h by 250% and 72 h by 170% in placebo group (Figure 4). DOMS remained significantly elevated in the spirulina group for 96h post exercise by 95.8%. No significant differences between groups were observed (*p* = 0.08; η^2^ = 0.127).

A repeated measures ANOVA did not reveal significant differences (*p* = 0.844; η^2^ = 0.018) between the two groups or time-dependent changes (*p* = 0.56; η^2^ = 0.014) for TAC (Figure 5). Finally, no significant differences (*p* = 0.419; η^2^ = 0.043) between the two groups or time-dependent changes (*p* = 0.222; η^2^ = 0.062) were found for protein carbonyls (PC) (Figure 6).

## 4. Discussion

To the best of our knowledge this was the first study that assessed the effects of spirulina supplementation on exercise performance, muscle damage and redox status following an acute eccentric exercise bout. The results indicate that supplementation with the particular dose (6 g per day) did not result in significant perturbations in indices of redox status, muscle damage or exercise performance compared to a placebo condition.

It is of great importance to understand how nutrition interacts with physiology in order to affect athletic performance. Comprehending the underpinning mechanisms that will aid in the alleviation of muscle damage and acceleration of the repair process will provide an insight into ways to help athletes recover quickly. Since spirulina apparently possesses antioxidant properties [6], this study aimed to provide an insight into the possible relationship between a nutritional supplement and recovery from a condition that muscle damage was evident.

Previous research indicates that acute muscle-damaging exercise may induce an increase in the production of ROS [26] primarily by phagocytes during the recovery process [27] and this perturbation in oxidative stress might play a potent role in the adaptation process following exercise-induced muscle damage [28]. There are many cases in sports where athletes participate in many games in a congested period of time (i.e., three games in a week) or take part in tournaments where they are involved in multiple games in a short period of time [24]. Finding ways to speed up the recovery from the temporal loss of muscle function and DOMS constitutes one of the goals of sports nutrition. Supplementations with antioxidants following a muscle-damaging exercise bout have shown equivocal results [26,29,30,31,32,33]. Supplementation with vitamin C [29], vitamin E [30] or a combination of the two [31] was effective in inhibiting muscle damage and delayed onset muscle soreness [33]. On the other hand, there were studies that showed that antioxidants did not affect exercise performance or muscle injury [32]. Furthermore, a combination of vitamin C and E supplementation did not affect exercise performance, redox status and muscle damage after one session of acute eccentric exercise [26]. In our study supplementation with spirulina, which is suggested to have antioxidant properties, was followed after a muscle damaging protocol to assess the secondary muscle damage and the associated oxidative stress and inflammation. The results indicate that no redox status or muscle damage and performance was evident. The inability to perturb the aforementioned parameters could be attributed to several factors. Dose, duration and timing of the supplement could be some of the reasons that explain the absence of perturbation in redox status. Furthermore, inter-individual variability in subjects’ redox status could be another factor for lack of changes. Recent studies have shown that greater improvements in exercise performance were seen in individuals with lower antioxidant levels at rest whereas performance was diminished in individuals with higher antioxidant levels at rest [34]. In addition, it has been shown that performing a repeated bout of muscle-damaging exercise results in lower oxidative stress and muscle damage [35]. Even though in this study the subjects that were included were screened to be individuals with no previous involvement with eccentric exercise in the previous 6 months, it is possible that some of them were trained prior to that with eccentric exercises and had a residual training effect that affected their response to eccentric exercise bout.

Previous work with spirulina supplementation immediately following a muscle-damaging exercise bout to assess the acute effects of supplementation is lacking. Lu et al. [17] used a short-term supplementation scheme with 7.5 g of daily dose for ten days and found lower levels of oxidative stress, as depicted by the lower levels of MDA and higher activity of GPX, and lower muscle damage, as indicated by the lower levels of LDH. Furthermore, greater exercise performance was reported since the subjects exercised more when they performed a Bruce incremental protocol [17]. However, in the aforementioned study, the nature of the exercise stimulus was different (walking vs. isokinetic exercise) than in the present study; they implemented a short supplementation period (3 weeks) prior to the performance of the exercise bout, the dose was different (7.5 vs. 6 gr) and the nature of exercise performance assessment (time to exhaustion vs. peak eccentric torque) was also different [17]. All or some of the above-mentioned reasons could explain the differences in the results between the two studies. Furthermore, in a recent study that evaluated the effects of 8 weeks of spirulina supplementation with different doses and assessed oxidative stress, muscle damage, inflammation and performance in rats revealed diminished oxidative stress and inflammation in animals receiving higher doses of spirulina [15]. It has to be pointed out that the above-mentioned study had a different methodological approach than our study since a prolonged supplementation period was followed and animals were used as subjects; as such, clear comparisons cannot be made. In addition, previous work from our lab indicates positive effects of 4-week spirulina supplementation in redox status and exercise performance [18]. Spirulina supplementation induced a significant increase in exercise performance, as was evident by increased time to exhaustion to a determined load, fat oxidation and reduced glutathione concentration and attenuated the exercise-induced increase in lipid peroxidation [18]. Nonetheless, a 4-week supplementation period preceded the exercise protocol which was a running protocol to exhaustion on the treadmill [17]. It is clear that the nature of the exercise stimulus of the aforementioned study is different than the one used in this study. Taken together, the results of the spirulina supplementation and exercise studies suggest that a supplementation period prior to an exercise that will assess exercise performance is needed in order to see positive results.

Limitations of this study include the lack of assessment of gender differences in the response to spirulina supplementation following such a kind of exercise protocol. Furthermore, this study assessed only the acute effects of spirulina supplementation and this consists of a limitation. Future studies could use a chronic supplementation protocol. Finally, another limitation in this study is the absence of inflammatory markers being assessed. Inflammation plays an important role in the exercise-induced muscle damage healing process [36] and undoubtedly information of such kind would strengthen the results of the study.

In conclusion, acute supplementation of spirulina following a muscle-damaging exercise protocol did not result in significant perturbations in redox status, muscle damage and exercise performance. Further studies with a prolonged supplementation period are needed to elucidate the effects of spirulina when intense exercise is performed.

## Figures and Tables

**Figure 1 ijms-22-03559-f001:**
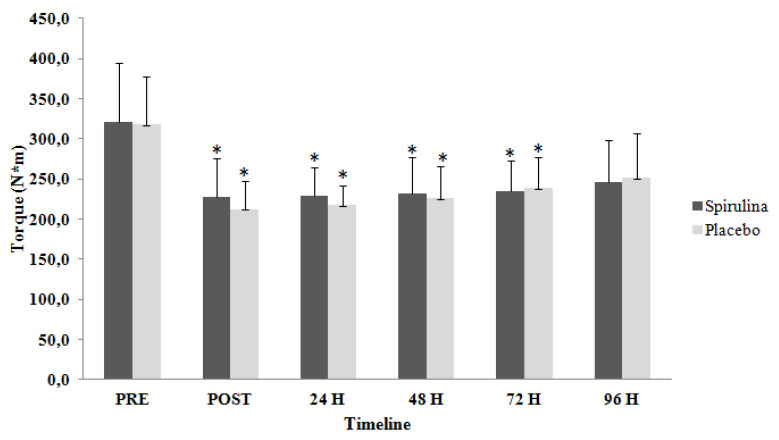
Change of eccentric peak torque for both limbs in response to eccentric protocol. PRE, Baseline measurement; H, Hours; * Significant difference with PRE.

**Figure 2 ijms-22-03559-f002:**
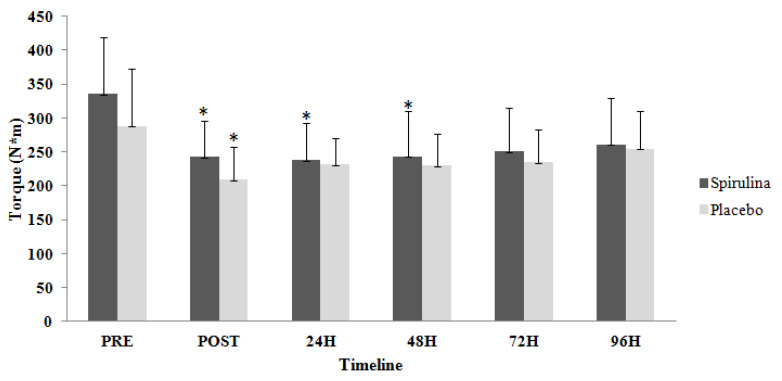
Change of eccentric peak torque for right limb in response to eccentric protocol. PRE, Baseline measurement; H, Hours; * Significant difference with PRE.

**Figure 3 ijms-22-03559-f003:**
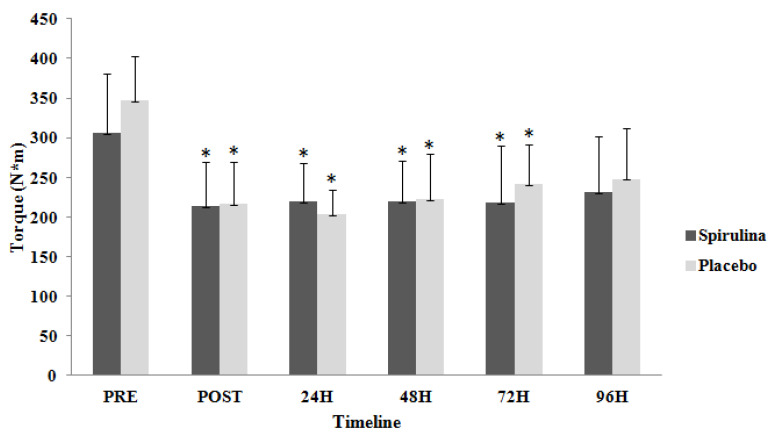
Change of eccentric peak torque for left limb in response to eccentric protocol. PRE, Baseline measurement; H, Hours; * Significant difference with PRE.

**Figure 4 ijms-22-03559-f004:**
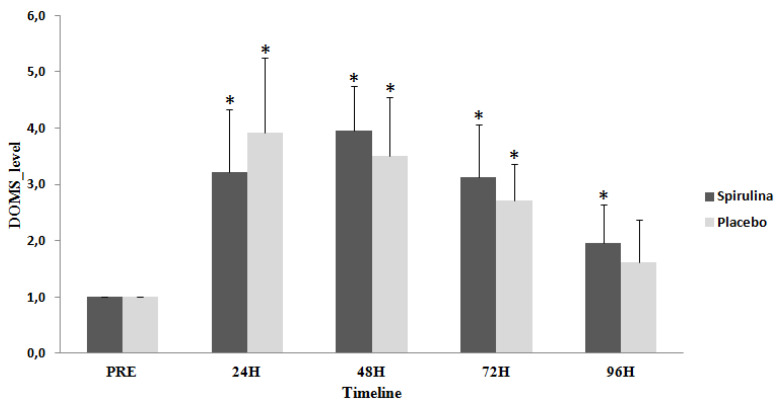
Change of Delay Onset Muscle Soreness (DOMS) in response to eccentric protocol. PRE, Baseline measurement; H, Hours; * Significant difference with PRE.

**Figure 5 ijms-22-03559-f005:**
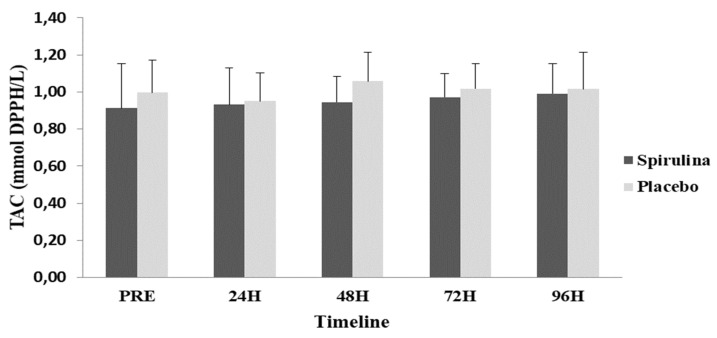
Change in total antioxidant capacity (TAC) concentration in response to eccentric protocol. PRE, Baseline measurement; H, Hours.

**Figure 6 ijms-22-03559-f006:**
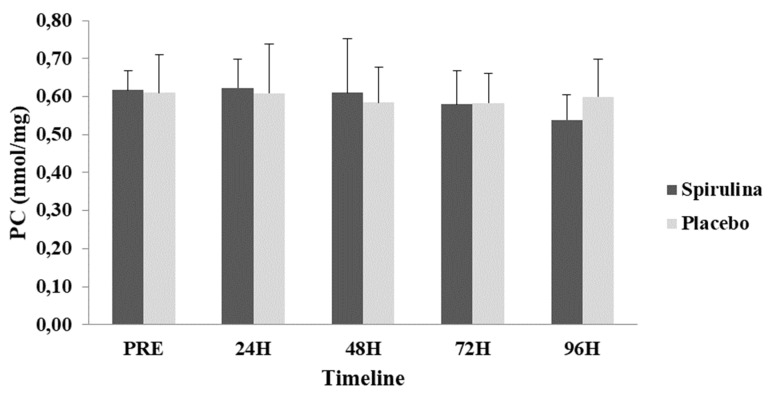
Change in protein carbonyls (PC) concentration in response to eccentric protocol. PRE, Baseline measurement; H, Hours.

**Table 1 ijms-22-03559-t001:** Personal characteristics of the participants.

Variable	Placebo	Spirulina
Age (years)	21.2 ± 2.2	22.5 ± 4.3
Weight (Kg)	73.5 ± 8.5	76.5 ± 7.7
Height (cm)	181.0 ± 5.9	179.8 ± 7.4
EPT_B_ (NM)	323.6 ± 71.0	320.9 ± 78.9
EPT_R_ (NM)	306.3 ± 74.7	335.6 ± 83.0
EPT_L_ (NM)	346.6 ± 56.3	306.3 ± 74.7

EPT_B_ (NM): eccentric peak torque for both legs; EPT_R_ (NM): eccentric peak torque for the right leg; EPT_L_ (NM): eccentric peak torque for the left leg.

## Data Availability

All data is contained within the article.

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
