# Peer review of "The Effects of Spirulina Supplementation on Redox Status and Performance Following a Muscle Damaging Protocol"

_ijms, 2021, doi:10.3390/ijms22073559_

Round 1

Reviewer 1 Report

The study carried out by Pappas et al. provides some information on the effect of spirulina supplementation on redox status during anaerobic strenuous exercise; in this sense, the concept is interesting. However, there are major issues that need to be addressed in order to improve the manuscript, especially regarding to Materials and Methods.

 Please, find below several comments:

  1. Materials and Methods: Participants

Authors indicate that twenty-four participants took part in the study; however, they do not describe the characteristics:

Were they only men, men and women? In this case, have the authors found any differences between men and women? The different physiology between genders may affect the final effect indeed. 

If so, the authors should indicate this information.

Were the participants well trained people, elite level, amateur?

I suggest adding this information

The sample size, (twenty-four participants, thats equals 12 people per group), is a very small sample size for a nutritional intervention / clinical study in humans. Please, reconsider and explain this aspect.

  1. Materials and Methods: supplementation

Regarding to the supplementation: the intervention group with spirulina supplementation (6 gr per day) vs placebo group.

The authors do not describe neither the specific composition, nor the form of supplementation (i.e capsules).  The composition profile for both groups should be identical except for the compound of interest, which in this case means, spirulina. Therefore, I suggest adding detailed information on the composition of both groups.

Also, how long was the supplementation period exactly?  It is not clearly described.

Was any method used to ensure that participants had the right amount of the product (spirulina) regarding to prescription/ guidelines?

Please, specify and indicate this information, as it would give a better understanding.

Spirulina supplementation was 6 gr/day, why did the authors use this specific amount?

I think the authors should consider mentioning this information and support it with evidence.

2.4 Blood collection

Regarding to the inflammatory response, authors could have taken advantage of the study, given the complexity of carrying out human studies, and measure inflammatory parameters (i.e. 8-hydroxy-2-deoxyguanosine (8-OHdG) or isoprostanes) which are directly related to the redox status during the physical activity performance, by collecting urine samples as well.

Please, reconsider and explain this aspect.

The authors should acknowledge the limitations of the study such as, small sample size, avoiding the physiological differences between men and women etc.

Again, and regarding to the inflammatory response, authors could have taken advantage of the study and measure inflammatory parameters (i.e. TNF-a, IL-6, IL-1) which are directly related to the redox status during the physical activity performance

In fact, in the conclusions, authors mention inflammatory parameters measured on previous studies using spirulina with positive results

Author Response

The study carried out by Pappas et al. provides some information on the effect of spirulina supplementation on redox status during anaerobic strenuous exercise; in this sense, the concept is interesting. However, there are major issues that need to be addressed in order to improve the manuscript, especially regarding to Materials and Methods.

 Please, find below several comments:

  1. Materials and Methods: Participants

Authors indicate that twenty-four participants took part in the study; however, they do not describe the characteristics:

Were they only men, men and women? In this case, have the authors found any differences between men and women? The different physiology between genders may affect the final effect indeed. 

If so, the authors should indicate this information.

Thank you for the suggestion to clarify this issue. Only male took part in this study and this information has been added in the methods section (2.1 Participants). Further information regarding the anthropometric and physiological characteristics of the participants appears on Table 1.

Were the participants well trained people, elite level, amateur?

I suggest adding this information

Thank you for your comment and suggestion. The participants were healthy, recreationally trained males. This info has been added in the manuscript in the methods section (2.1 Participants)

The sample size, (twenty-four participants, thats equals 12 people per group), is a very small sample size for a nutritional intervention / clinical study in humans. Please, reconsider and explain this aspect.

Thank you for your comment and suggestion. Prior to the study a power analysis was performed to identify the number of participants. That is explained in the methods section (2.1 Participants).

A power analysis was conducted by using the G*Power version 3.1.9.2 program. It was estimated that a total sample of 24 participants was necessary to identify statistically meaningful trial effects (with a probability error of 0.05, a statistical power of 80% and an effect size of 0.25). 

  1. Materials and Methods: supplementation

Regarding to the supplementation: the intervention group with spirulina supplementation (6 gr per day) vs placebo group.

The authors do not describe neither the specific composition, nor the form of supplementation (i.e capsules).  The composition profile for both groups should be identical except for the compound of interest, which in this case means, spirulina. Therefore, I suggest adding detailed information on the composition of both groups.

Thank you for the suggestion. A dietitian was responsible for constructing the capsules. The daily dosage of spirulina (Algae AC, Serres, Greece) was 6 gr per day and the capsules (1 gr each) were taken three times per day before meals. Spirulina was given in a capsule form. Placebo was also given in a capsule that contained wheat flour and was of the same color and size as spirulina’s capsule to avoid identification of the supplement. Furthermore, subjects were also asked to return at the end of trial the capsules that would not be consumed. There was not a subject that returned capsules back. The dosage of 6 gr per day was chosen because it is the same with the one used by our group in a previous study (Kalafati et al. 2010) and similar to the ones used by relevant human studies (7.5 gr per day) (Lu et al. 2006).

This information was added in the methods section (2.1 Participants)

Also, how long was the supplementation period exactly?  It is not clearly described.

Thank you for your comment and suggestion. We added this information at the methods section. (2.3 eccentric exercise protocol – final sentence).

After the end of the eccentric exercise protocol subjects were provided with the spirulina or placebo capsules and they were obliged to take 6 capsules a day at the previously mentioned time points until the final day of data collection.

Was any method used to ensure that participants had the right amount of the product (spirulina) regarding to prescription/ guidelines?

Please, specify and indicate this information, as it would give a better understanding.

Spirulina supplementation was 6 gr/day, why did the authors use this specific amount?

I think the authors should consider mentioning this information and support it with evidence.

Thank you for your comment. The dosage of 6 gr per day was chosen because it is the same with the one used by our group in a previous study (Kalafati et al. 2010) and similar to the ones used by relevant human studies (8 gr per day) (Lu et al. 2006).

2.4 Blood collection

Regarding to the inflammatory response, authors could have taken advantage of the study, given the complexity of carrying out human studies, and measure inflammatory parameters (i.e. 8-hydroxy-2-deoxyguanosine (8-OHdG) or isoprostanes) which are directly related to the redox status during the physical activity performance, by collecting urine samples as well.

Please, reconsider and explain this aspect.

 Thank you for the suggestion. Indeed, assessing 8-OHdG or isoprostanes would provide additional and important information. Unfortunately, no urine samples were collected. Nevertheless, this is a point of consideration for future research.

The authors should acknowledge the limitations of the study such as, small sample size, avoiding the physiological differences between men and women etc.

Thank you for the suggestion. Limitations were acknowledged and appear at the end of the discussion section

Limitations of this study include the lack of assessment of gender differences in the response to spirulina supplementation following such a kind of exercise protocol. Furthermore, this study assessed only the acute effects of spirulina supplementation and this consists of a limitation. Future studies could use a chronic supplementation protocol. Finally, another limitation in this study is the absence of inflammatory markers being assessed. Inflammation plays an important role in the exercise-induced muscle damage healing process and undoubtedly information of such kind would strengthen the results of the study.

Again, and regarding to the inflammatory response, authors could have taken advantage of the study and measure inflammatory parameters (i.e. TNF-a, IL-6, IL-1) which are directly related to the redox status during the physical activity performance

In fact, in the conclusions, authors mention inflammatory parameters measured on previous studies using spirulina with positive results

Thank you for the suggestion. We do agree that assessing inflammatory markers would add information about possible mechanisms being modulated by the supplement.  Unfortunately, there is no blood samples left to perform analysis of additional variables. This is highlighted as a limitation of the study as explained in the previously mentioned point.

Reviewer 2 Report

Dear authors,

I appreciate this study where it has been studied the possible effect of one of the supplements that have increased its consume on the athlete population (spirulina) on an eccentric training session and the response post-exercise. Although the topic could be of interest for athletic population, authors haven´t specified a good cause-effect relation between the variables of the study. In addition, authors present a limited knowledge about the physiology assessment and the study approach an objective that it´s not good stablished. Also, the soundness is so poor and the methods (including statistical treatment) is very poor. Therefore, this manuscript cannot be accepted for publishing.

Authors says the next: “Previous work on spirulina supplementation and exercise training has revealed positive results [15-18] and work from our laboratory has shown that acute supplementation of spirulina increased aerobic exercise performance, decreased carbohydrate and increased fat oxidation and lower lipid peroxidation was evident [19]”. Undoubtedly, authors are wrong because this interpretation of the term “exercise performance” is wrong. An athlete increases its athletic performance when need less time for covering a distance or increase the time to exhaustion to a determined load. To have a less VO2 at a submaximal load (economy) could be a good metabolic parametric and be associated with a hypothetical enhancement of performance during an endurance test. Nevertheless, it´s not necessary that a metabolic advantage imply a sport performance enhancement.

Based on the previous commentary, authors must make a new contextualization. In addition, authors don´t explain the relation between the mechanism action of spirulina that could act as an ergogenic aid. Why spirulina could enhance the performance during an anaerobic exercise (eccentric contractions) and DOMS? Based on the introduction, there isn´t exist any reason that justify the objective of the study.

In addition, to the incorrect concept of the authors about sport performance, the confusion is increased when it´s used MET as maximum eccentric peak torque is a term very confused. Authors need to know that in physiology and Sport Sciences, MET (metabolic equivalent of task) is associated with a VO2 corresponding to 3.5 ml/kg/min.

At methodological level, I´m very worried about the placebo that authors haven´t clarified. What was the placebo used?

On methods section is not explained that the exercise was performed with both and unilateral leg. I have surprised when I have read it on Table 1.

On the treatment of the data, I cannot understand why authors haven´t analysed the interaction time·supplementation. This interaction as the post-hoc of the interaction is the information more interesting and correct for answering to your objectives. In addition, the effect sizes it´s recommendable add it.  

Regarding to the discussion, it´s not suitable because is not discussed the mechanism of the effect of spirulina with an eccentric training session.

Author Response

I appreciate this study where it has been studied the possible effect of one of the supplements that have increased its consume on the athlete population (spirulina) on an eccentric training session and the response post-exercise. Although the topic could be of interest for athletic population, authors haven´t specified a good cause-effect relation between the variables of the study. In addition, authors present a limited knowledge about the physiology assessment and the study approach an objective that it´s not good stablished. Also, the soundness is so poor and the methods (including statistical treatment) is very poor. Therefore, this manuscript cannot be accepted for publishing.

Authors says the next: “Previous work on spirulina supplementation and exercise training has revealed positive results [15-18] and work from our laboratory has shown that acute supplementation of spirulina increased aerobic exercise performance, decreased carbohydrate and increased fat oxidation and lower lipid peroxidation was evident [19]”. Undoubtedly, authors are wrong because this interpretation of the term “exercise performance” is wrong. An athlete increases its athletic performance when need less time for covering a distance or increase the time to exhaustion to a determined load. To have a less VO2 at a submaximal load (economy) could be a good metabolic parametric and be associated with a hypothetical enhancement of performance during an endurance test. Nevertheless, it´s not necessary that a metabolic advantage imply a sport performance enhancement.

Thank you for your comment. We agree that enhancement in aerobic exercise performance can be evaluated by covering a specified distance with less time or increasing the time to exhaustion to a determined load. In the statement that the reviewer is referring to and appears on the introduction section of our manuscript we cite our previous work from our laboratory (Kalafati et al. 2010) where subjects were able to run longer following a spirulina supplementation protocol compared to a placebo condition. We rephrased the sentence by adding a clarification remark and now the sentence reads

Previous work on spirulina supplementation and exercise training has revealed positive results [15-18] and work from our laboratory has shown that acute supplementation of spirulina increased aerobic exercise performance, as it was evident by increased time to exhaustion to a determined load, decreased carbohydrate and increased fat oxidation and lower lipid peroxidation was evident [19]”.

Based on the previous commentary, authors must make a new contextualization. In addition, authors don´t explain the relation between the mechanism action of spirulina that could act as an ergogenic aid. Why spirulina could enhance the performance during an anaerobic exercise (eccentric contractions) and DOMS? Based on the introduction, there isn´t exist any reason that justify the objective of the study.

Thank you for the comment and the suggestion. We added a new paragraph (paragraph 4) in the introduction section explaining why supplementing an athlete with spirulina could potentially affect DOMS and speed up the recovery of muscle function and why this is important for an athletic performance.

In addition, to the incorrect concept of the authors about sport performance, the confusion is increased when it´s used MET as maximum eccentric peak torque is a term very confused. Authors need to know that in physiology and Sport Sciences, MET (metabolic equivalent of task) is associated with a VO2 corresponding to 3.5 ml/kg/min.

Thank you for the suggestion. MET has been changed to torque throughout the manuscript.

At methodological level, I´m very worried about the placebo that authors haven´t clarified. What was the placebo used?

Spirulina was given in a capsule form. Placebo was also given in a capsule that had the same color and size to avoid identification of the supplement. Subjects could not distinguish which supplement was the one containing the spirulina and furthermore they were asked to return the capsules that were not consumed at the end of trial. There was not a subject that returned capsules back.

This information was added in the methods section (2.1 Participants)

On methods section is not explained that the exercise was performed with both and unilateral leg. I have surprised when I have read it on Table 1.

Thank you for the comment. We added this information on the methods section (2.2 Experimental Design)

On the treatment of the data, I cannot understand why authors haven´t analysed the interaction time·supplementation. This interaction as the post-hoc of the interaction is the information more interesting and correct for answering to your objectives. In addition, the effect sizes it´s recommendable add it.  

Thank you for the comment. We added this information on the results section (3. Results)

Regarding to the discussion, it´s not suitable because is not discussed the mechanism of the effect of spirulina with an eccentric training session.

Thank you for the comment. Potential mechanisms by which spirulina could influence the assessed variables in this study have been added in the discussion

Round 2

Reviewer 1 Report

The manuscript has been improved regarding to the Materials and Methods section, by providing additional data on participants characteristics, supplementation and sample collection.

However, the authors could have taken advantage of the study by doing a wider analysis to improve the quality of the results and the study overall. The article could be rewritten from a more useful point of view for clinicians.

Author Response

The manuscript has been improved regarding to the Materials and Methods section, by providing additional data on participants characteristics, supplementation and sample collection.

However, the authors could have taken advantage of the study by doing a wider analysis to improve the quality of the results and the study overall. The article could be rewritten from a more useful point of view for clinicians.

We would like to thank the reviewer for the positive comments.

We agree with the reviewer that another direction could be given to the manuscript if a meaningful change was found due to spirulina supplementation. However, we believe that this is quite difficult to do since there were not any statistical differences due to spirulina supplementation. In addition, the purpose of our study was to assess the exercise-induced effects on redox status and muscular function and not on clinical health issues.

Reviewer 2 Report

Dear authors,

I lament that authors maintain the same wrong conceptualization about endurance performance. Authors have answered to me that have solved my commentaries and author have included the next information: “Previous work on spirulina supplementation and exercise training has revealed positive results [15-18]”.

I have read every reference. Therefore, I attached them as supplementary files. In these files can be checked the next:

  • Reference 15 is a study carry out on rat and don´t include any performance test.
  • Reference 16 included an incremental test to fatigue as performance test and it didn´t observe any statistical effect of the supplementation.
  • Reference 17 include a wrong statistical treatment. Although the statistical treatment is incorrect for detecting a possible effect of the supplementation (authors should be performed an ANOVA for detecting the interaction time·supplementation), in table 3 can be observed any effect after exercise and non-exercise program on both, supplementation and placebo condition.
  • Reference 18 only include biochemical parameters and don´t include any sport performance assessment.

Therefore, only the reference 19 has observed a statistical effect of spirulina on endurance performance. I believe that authors have mistaken in its contextualization and I maintain that authors don´t differentiate correctly between sport performance variables and physiological responses to exercise. In anyway, the mistakes that I have identified are very serious.

Regarding to my previous commentaries about the absence of contextualization of the eccentric training, authors have included the next: “Anaerobic and unaccustomed eccentric exercise has been shown to result in ultrastructural muscular disruption, delayed onset muscle soreness (DOMS), enhancement of oxidative stress and reductions in performance related variables i.e. impaired muscle force production [15, 16]”. It´s different an unaccustomed eccentric exercise and an anaerobic exercise. For example, a swimmer could swim a session consisting on 10x50 m sprint that it´s an anaerobic effort, but it will not originate muscular disruptions. Once again, authors don´t understand correctly the exercise physiology and reflex a poor awareness on the exercise physiology response during exercise and it´s concepts.

In general, it´s necessary to clarify that an eccentric strength training is a non-representative example of the anaerobic efforts. Therefore, the contextualization of the introduction is so poor.

I appreciate that authors have extended explanation related to the placebo elaboration, but it´s impossible to affirm “Subjects could not distinguish which supplement was the one containing the spirulina and furthermore they were asked to return the capsules that were not consumed at the end of trial”. For doing this affirmation, authors should question to the subject it they believed that they ingested placebo or spirulina. I believe that this analysis is not important, but it´s the correct methodology for can include this affirmation.

Regarding to the results section, the information presented is very confused. I have doubts about the ANOVA because it´s seem a pairwise comparison and not an ANOVA of two ways and repeated measured. In addition, in this type of ANOVA is required a partial eta squared (authors have included a Cohen´s d).

Regarding to the discussion, all the information is not endorsed by any reference. Therefore, like all the manuscript, it´s very speculative and present a very low scientific soundness. This very low scientific soundness, misinterpretation of the physiological response to exercise and the incorrect statistical treatment of the data make that this article cannot be published in any scientific journal. Therefore, this manuscript must be rejected.

Author Response

Dear authors,

I lament that authors maintain the same wrong conceptualization about endurance performance. Authors have answered to me that have solved my commentaries and author have included the next information: “Previous work on spirulina supplementation and exercise training has revealed positive results [15-18]”.

I have read every reference. Therefore, I attached them as supplementary files. In these files can be checked the next:

  • Reference 15 is a study carry out on rat and don´t include any performance test.
  • Reference 16 included an incremental test to fatigue as performance test and it didn´t observe any statistical effect of the supplementation.
  • Reference 17 include a wrong statistical treatment. Although the statistical treatment is incorrect for detecting a possible effect of the supplementation (authors should be performed an ANOVA for detecting the interaction time·supplementation), in table 3 can be observed any effect after exercise and non-exercise program on both, supplementation and placebo condition.
  • Reference 18 only include biochemical parameters and don´t include any sport performance assessment.

Therefore, only the reference 19 has observed a statistical effect of spirulina on endurance performance. I believe that authors have mistaken in its contextualization and I maintain that authors don´t differentiate correctly between sport performance variables and physiological responses to exercise. In anyway, the mistakes that I have identified are very serious.

The statement that appears in the introduction section of our manuscript states that “…Previous work on spirulina supplementation and exercise training has revealed positive results..” and Brito et al. [15 in our reference list] study revealed that spirulina supplementation improved the antioxidant capacity compared to a control condition in exercised rats and to our opinion this is a positive result when someone assess the effects of spirulina supplementation. We did not specify the improvement in regards to exercise performance.

Gurney & Spendiff  [16 in our reference list] examined the effects of spirulina supplementation and found lower oxygen consumption during submaximal exercise and higher oxygen uptake by 8.9% at the point of fatigue when spirulina was given as a supplement compared to a control condition. This again to our view is a positive result. We do agree with the reviewer that performance wise there were not significant differences but the higher oxygen consumption to fatigue achieved by the subjects is indeed a positive result.

Hernandez-Lepe et al. [17 in our initial submission reference list] study indeed used a wrong statistical method and we eliminated it from the manuscript.

Lu et al. study [18 in our reference list] examined the spirulina supplementation and found higher glutathione peroxidase and lower lactate dehydrogenase levels which both constitute positive changes. Taken all the above collectively we believe that our statement that “..Previous work on spirulina supplementation and exercise training has revealed positive results..” is correct and by no means we reported incorrect and misleading information.

Regarding to my previous commentaries about the absence of contextualization of the eccentric training, authors have included the next: “Anaerobic and unaccustomed eccentric exercise has been shown to result in ultrastructural muscular disruption, delayed onset muscle soreness (DOMS), enhancement of oxidative stress and reductions in performance related variables i.e. impaired muscle force production [15, 16]”. It´s different an unaccustomed eccentric exercise and an anaerobic exercise. For example, a swimmer could swim a session consisting on 10x50 m sprint that it´s an anaerobic effort, but it will not originate muscular disruptions. Once again, authors don´t understand correctly the exercise physiology and reflex a poor awareness on the exercise physiology response during exercise and it´s concepts.

In general, it´s necessary to clarify that an eccentric strength training is a non-representative example of the anaerobic efforts. Therefore, the contextualization of the introduction is so poor.

By no means we put anaerobic exercise with the unaccustomed eccentric exercise in the same category. That is evident by the title of the manuscript where we clearly indicate that the exercise protocol that was followed in our design was a muscle damaging one. If you search in our manuscript for the word “anaerobic” you will get 4 results, one of which appears in the literature. This clearly indicates that the focus of our manuscript has nothing to do with anaerobic exercise. Undoubtedly though there are reports in the literature indicating that anaerobic exercise can result in elevated markers of muscle damage (Hammuda et al. 2012) and oxidative stress (Bloomer et al. 2005). However, since the focus of our manuscript is not on the anaerobic exercise but on the unaccustomed eccentric exercise we did not include them in the manuscript. Furthermore, we eliminated the term “anaerobic” from the beginning of the sentence to make our statement more robust and focused on the unaccustomed eccentric exercise.

I appreciate that authors have extended explanation related to the placebo elaboration, but it´s impossible to affirm “Subjects could not distinguish which supplement was the one containing the spirulina and furthermore they were asked to return the capsules that were not consumed at the end of trial”. For doing this affirmation, authors should question to the subject it they believed that they ingested placebo or spirulina. I believe that this analysis is not important, but it´s the correct methodology for can include this affirmation.

It is difficult for us to follow the reviewer’s comment but is common practice in sports nutrition to produce capsules of the same color and texture so subjects could not distinguish the supplement and ask them to return the supplement so you can establish the adherence rate. In case that the reviewer implies that a statement should be added in the methods section we added the following one.

“Subjects were asked whether they could distinguish between supplements and their response was that could not identify the spirulina or placebo supplement.”  

Regarding to the results section, the information presented is very confused. I have doubts about the ANOVA because it´s seem a pairwise comparison and not an ANOVA of two ways and repeated measured. In addition, in this type of ANOVA is required a partial eta squared (authors have included a Cohen´s d).

The design of the experiment was a 2X2 (group X time) and we believe that the correct statistical analysis is a 2-way ANOVA. Following to the reviewer’s suggestion for effect size, partial eta squared are reported.

Regarding to the discussion, all the information is not endorsed by any reference.

Additional references have been included now in the manuscript in the discussion section.

Therefore, like all the manuscript, it´s very speculative and present a very low scientific soundness. This very low scientific soundness, misinterpretation of the physiological response to exercise and the incorrect statistical treatment of the data make that this article cannot be published in any scientific journal. Therefore, this manuscript must be rejected.

We obviously disagree with the reviewer since we thoroughly explained our results from a physiological point of view giving emphasis on the effects of the supplement on exercise-induced muscle damage and redox dependent changes. We believe that the results are novel and will add in the existing body of literature.
